# The Trivalent Recombinant Chimeric Proteins Containing Immunodominant Fragments of *Toxoplasma gondii* SAG1 and SAG2 Antigens in Their Core—A Good Diagnostic Tool for Detecting IgG Antibodies in Human Serum Samples

**DOI:** 10.3390/ijms26125621

**Published:** 2025-06-12

**Authors:** Bartłomiej Tomasz Ferra, Maciej Chyb, Marta Skwarecka, Justyna Gatkowska

**Affiliations:** 1Department of Tropical Parasitology, Institute of Maritime and Tropical Medicine in Gdynia, Medical University of Gdańsk, Powstania Styczniowego 9B, 81-519 Gdynia, Poland; 2Department of Molecular Microbiology, Faculty of Biology and Environmental Protection, University of Lodz, Banacha 12/16, 90-237 Łódź, Poland; maciej.chyb@edu.uni.lodz.pl (M.C.); justyna.gatkowska@biol.uni.lodz.pl (J.G.); 3Bio-Med-Chem Doctoral School of the University of Lodz and Lodz Institutes of the Polish Academy of Sciences, Faculty of Biology and Environmental Protection, University of Lodz, Banacha 12/16, 90-237 Łódź, Poland; 4Department of Molecular Genetics of Bacteria, Faculty of Biology, University of Gdansk, Wita Stwosza 59, 80-308 Gdańsk, Poland; marta.skwarecka@ug.edu.pl

**Keywords:** *Toxoplasma gondii*, toxoplasmosis, recombinant chimeric protein, human serum samples, ELISA, serodiagnosis, antibodies

## Abstract

Toxoplasmosis is one of the most common and neglected parasitic diseases caused by the intracellular parasite *Toxoplasma gondii*. The parasitic invasion in humans is associated with certain problems, such as the lack of effective immunoprophylaxis or a complex diagnostic algorithm, which require continuous improvement. Both problems can be overcome by the recent development of *T. gondii* proteomics, which has allowed the design of different recombinant antigens. In this study we evaluated the potential usefulness of nineteen recombinant chimeric *T. gondii* proteins for serodiagnosis. A chimeric antigen composed of the surface antigens SAG1-SAG2 was developed and used as the basis for the generation of 18 subsequent trivalent chimeric antigens containing different immunodominant fragments of the parasite proteins. The recombinant antigens were used in an indirect enzyme-linked immunosorbent assay (ELISA) test to evaluate their ability to detect specific IgG antibodies in human sera. A total of 338 human sera were analyzed to assess the sensitivity and specificity of the tests. Sixteen of the antigens tested demonstrated 100% sensitivity and specificity in the ELISA for the detection of specific IgG antibodies. These results provide an optimistic outlook for the potential replacement of the currently used native antigen mix with recombinant antigens in human *T. gondii* serodiagnostics.

## 1. Introduction

*Toxoplasma gondii* is a highly successful parasite that has capacity to infect a wide range of warm-blooded animals, including humans [1]. The parasite’s ability to persist in the host for a lifetime and its wide distribution contribute to its global prevalence. It is estimated that approximately one-third of the world’s human population is infected with the parasite. The transmission routes of *T. gondii* to humans and animals have been the focus of extensive research in recent years [2]. Felines play a pivotal role as the definitive hosts of the parasite, where young animals typically acquire infection for the first time by ingesting the tissue cyst of the parasite contained within food (e.g., infected rodents or birds, raw meat with tissue cysts, etc.). Following ingestion, the parasites penetrate the enteroepithelial cells of the cat’s intestine, thereby initiating the sexual cycle and forming oocysts. These oocysts are then excreted in the cat’s feces for a period of 5 to 14 days after infection. Intermediate hosts, including humans and animals, can become infected by consuming sporulated oocysts in contaminated food and water [3]. In particular, research has indicated that toxoplasmosis is a major cause of mortality in certain wildlife species, such as the California sea otter, which is thought to be due to oocyst contamination in the oceans [4,5]. Asexual multiplication by tachyzoites occurs within various host cells and leads to the formation of tissue cysts containing bradyzoites. The consumption of raw or undercooked meat from animals containing tissue cysts poses a significant risk of infection to humans [3]. Research has indicated that meat from pigs, sheep, and goats harbours the highest risk of infection, followed by free-range poultry and game animals [6,7,8]. Furthermore, transmission can occur through the consumption of milk from infected animals or vertically from the mother to the fetus during pregnancy [3]. Furthermore, the transplantation of organs from an infected donor, especially in conjunction with immunosuppressive treatment, also harbours the risk of transmission to humans [3]. Although infection with *T. gondii* generally manifests in mild clinical symptoms, there are exceptions to this general pattern both between and within different host species [9]. It is widely acknowledged that pregnant women and immunocompromised individuals are considered the main risk groups for severe complications of *T. gondii* infection. However, the infection can also lead to ocular disease and psychiatric disorders [9,10,11]. In animals, especially sheep, goats, and pigs, *T. gondii* infection can lead to abortions and congenital infections, resulting in significant economic losses for the farming industry.

The primary strategies to control the parasite focus on educating people about transmission routes to prevent infection, especially in the context of pregnant women. Drug treatments, such as spiramycin, pyrimethamine, and sulfadiazine, are used in cases of congenital infection or to prevent reactivation in immunocompromised individuals. However, these drugs have limited efficacy against the tissue cysts stage of the parasite, and there is a possibility of adverse effects, particularly in pregnant women [9]. Given the challenges associated with current treatment options, vaccination is a promising alternative to combat *T. gondii* infection. Currently, the only commercially available vaccine against *T. gondii* is ‘Toxovax’, which is intended for the vaccination of sheep. However, this vaccine is not suitable for use in humans because it contains a live attenuated S48 strain of *T. gondii*, which may raise safety concerns related to a return to the invasive form [12]. Considering the lack of an effective vaccine that can be utilized across a wide range of parasite hosts and the lack of drugs with minimal side effects, we should focus on the diagnosis of parasite infection, which, unfortunately, also causes additional problems. The proper diagnosis of *T. gondii* infection and the phase of the disease is particularly important in the case of pregnant women, fetuses, newborns with congenital toxoplasmosis, and in people with immunodeficiencies. The tests used for this purpose are mainly based on the determination of the titre of IgM and IgG antibodies and the avidity of IgG antibodies [13,14,15,16,17]. The initial antibodies which appear in the bloodstream during the second week after infection are IgM antibodies, the titre of which rises over the next four weeks. These antibodies persist for about four months and then disappear. The presence of antibodies of this class indicates the acute phase of toxoplasmosis in most cases. However, in some patients, IgM immunoglobulins may persist for much longer. IgG antibodies appear in circulation approximately 2–3 weeks after infection, and their titre rises rapidly over the next 2 months. Thereafter, the concentration of IgG antibodies slowly decreases over several months, and then the low titre usually persists for the rest of life. The correlation between the titres of IgM and IgG antibodies can, in some cases, be used to determine the phase of the disease, e.g., a positive result for IgM antibodies and an increase in IgG antibodies in subsequent tests indicates primary invasion of the parasite. However, in most cases, further testing is necessary to determine whether IgA and/or IgE antibodies are present, and/or to perform an IgG avidity test. The IgA and IgE antibodies appear in the body at about the same time as IgM antibodies, but their detection can be very difficult. Therefore, in the routine diagnosis of toxoplasmosis the avidity of IgG antibodies is usually tested to determine the binding strength of antigen–antibody complexes, which is something that increases with the development of the disease. The IgG avidity index in serum increases over time from the moment of the *T. gondii* invasion. Low avidity indicates the initial, active phase of infection. The switch from low to high avidity takes about 12-16 weeks. A high IgG avidity index indicates that the infection occurred more than four months ago. This method makes it possible to clearly determine the phase of the *T. gondii* infection in relation to the results of tests for the presence of IgM and IgG antibodies [13,14,15,16,17].

Currently, serological laboratories employ commercially available diagnostic tests based on native *Toxoplasma* lysate antigen (TLA) to detect specific anti-*T. gondii* antibodies [18]. However, the use of this antigen preparation is associated with several issues and drawbacks. The production of TLA requires continuous culture of the parasite, which can be costly and labour intensive. The antigenic composition of TLA can vary depending on the culture conditions and the efficacy of parasite cell disruption. This variability necessitates the standardization of each batch of TLA for uniform diagnostic performance, which can present certain challenges [18]. TLA consists of numerous native antigens, making it difficult to fully characterize, and this complexity can compromise the diagnostic accuracy of TLA-based assays. TLA does not allow differentiation between acute and chronic phases of toxoplasmosis in a single immunoassay [19]. This distinction is particularly critical for pregnant women as it impacts clinical management and treatment decisions. Research carried out so far has led to the discovery and identification of numerous parasite antigens [20]. Furthermore, understanding the antigenic structure of the parasite has enabled the development of many recombinant proteins that may be used as an alternative to TLA in the serodiagnosis of *T. gondii* invasion in humans and animals. The use of recombinant proteins offers several advantages over TLA for the diagnosis of *T. gondii* infection. Recombinant proteins can be produced relatively inexpensively, making them more accessible for diagnostic purposes. Recombinant proteins are fully defined and can be more easily standardized compared to TLA, resulting in improved consistency and reliability in diagnostic assays. Recombinant proteins can be selected to target specific stages of *T. gondii* infection, allowing differentiation between acute and chronic phases [19]. This specificity increases the precision of diagnostic tests, thereby facilitating more accurate clinical decision-making. In general, the use of recombinant proteins offers a promising solution to overcome the limitations associated with TLA-based serological diagnostics for *T. gondii* infection. By providing a cost-effective, reliable, and specific diagnostic tool, recombinant proteins have the potential to improve patient care and management, particularly for vulnerable populations such as pregnant women.

The aim of this study was to evaluate the diagnostic utility of 18 newly produced recombinant trivalent chimeric proteins (all containing the same immunodominant fragment of the SAG1 and SAG2 *T. gondii* antigens, and an additional immunodominant fragment of one of the parasite antigens, such as AMA1, GRA1, GRA2, GRA5, GRA6, GRA7, GRA9, LDH2, MAG1, MIC1, MIC3, P35, and ROP1) for the detection of specific anti-*T. gondii* IgG antibodies in human serum samples. Antigens associated with the recognition, attachment, invasion, colonization, and multiplication of *T. gondii* in host cells have been selected for the construction of recombinant trivalent chimeric proteins with potential diagnostic significance [21,22,23,24,25,26,27,28,29,30,31]. These antigens belong to the following four main families of parasite antigens: (a) Surface antigens (SAGs): This family includes SAG1, SAG2, and P35, which are part of the SAG1-related sequence proteins (SRSs) family and are crucial for the initial attachment of the parasite to host cells, interaction with the host immune response, and the modulation of parasite virulence; (b) Rhoptry antigens (ROPs): ROP1 is an example of this family. ROP proteins are secreted during the invasion process and play an essential role in facilitating host cell membrane folding, the formation of parasitophorous vacuoles (PVs), and other essential cellular events during invasion [32]; (c) Microneme antigens (MICs): MIC1, MIC3, and AMA1 are representative antigens of this family. MICs are secreted from micronemes and assist in adhesion to host cells, disrupting the host cell membrane to allow parasite penetration. They form adhesive complexes that facilitate targeted adhesion to surface receptors in host cells [33]; (d) High-density granule antigens (GRAs): GRA1, GRA2, GRA5, GRA6, GRA7, and GRA9 are examples of this family. GRA proteins, secreted by high-density granule vesicles, play a crucial role in the modification of the parasitophorous vacuole (PV) and are involved in the growth and development of *T. gondii* within the vacuole [34,35]. Furthermore, two antigens that do not belong to the above-mentioned families were selected for the study, namely lactate dehydrogenase (LDH2) and the major matrix antigen of the tissue cyst (MAG1).

## 2. Results

### 2.1. Plasmids Construction, Expression, and Purification of Recombinant Chimeric Proteins

For all 19 recombinant chimeric proteins used in this study, the previously described [36] and the newly constructed SS-GRA5S recombinant plasmids containing fusion genes were obtained by genetic engineering and molecular biology methods (Appendix A). The correctness of the cloning was confirmed by DNA sequencing (Genomed S.A., Warsaw, Poland). High expression levels of the fusion genes were observed in *Escherichia coli* Rosetta(DE3)pLysS. The recombinant chimeric proteins were expressed as insoluble proteins. The calculated molecular mass of the recombinant chimeric proteins ranged from 50.15 to 106.08 kDa. The recombinant chimeric proteins were purified by metal affinity chromatography under denaturing conditions with 5 M urea. Details of the properties of the produced recombinant chimeric proteins can be found in Appendix A. The expression system used allows for the recovery of 18–35 mg of purified proteins per litre of bacterial culture. The purification led to an electrophoretically homogeneous preparation with a purity of over 90%.

### 2.2. Immunoreactivity of Human Anti-T. gondii Antibodies in the ELISA Test

The tested antigens were used in an indirect ELISA test to determine the reactivity of the IgG class of human sera. The sera were categorized into the following five groups: I—uninfected, II—suspected acute, IIIA—chronic, IgG > 200 IU/mL, IIIB—chronic, IgG 101–200 IU/mL, and IIIC—chronic, IgG < 100 IU/mL (Figure 1).

The reactivity of IgG antibodies with TLA was found to be particularly high in group IIIA, i.e., in the chronic phase with the highest titre, and gradually decreased in groups IIIB and IIIC as the titre decreased. The distributions of test values for the chronic group IIIC are similar to those of the acute phase (group II). The values for each serum sample used for every new recombinant chimeric antigen are similar to the test value obtained with TLA. The validity of this observation is supported by the results of the Pearson correlation analysis, as shown in Table 1 and Figure 2. The correlation analysis was visualized using a scatter plot with line regression fit (Appendix A). The results of the individual sera show a significant correlation with the values archived by using TLA, with R^2^ values ranging from 0.78 to 0.89. A similar result is obtained when analyzing the correlation of the trivalent proteins’ values with the SAG1-SAG2 antigen, with a significant correlation in the range of R^2^ values of 0.78 to 0.94 (Table 1/Appendix A).

A receiver operating characteristic (ROC) analysis was performed to evaluate the diagnostic efficacy of the tests. This analysis revealed the cut-off value and the associated sensitivity and specificity of the test for each antigen. The results are shown in Table 2 and Figure 3.

The ROC analysis of the TLA used in current diagnostic procedures showed a sensitivity of 100% (CI 95% 98.12–100) and a specificity of 100% (CI 95% 97.29–100), with a cut-off value of 0.7245. Using the studied recombinant chimeric *T. gondii* antigens the tests had different cut-off values for each of the proteins, with each of the tests having a specificity of 100% and sensitivity ranging from 98.5 to 100%. It should be noted that most antigens showed a sensitivity of 100%, and only three (SS-ROP1, -GRA6, -GRA1) showed lower values. The lowest sensitivity was found for the SS-GRA1 antigen (95% CI 95.68–99.59%).

To compare the results obtained for the individual recombinant chimeric antigens, the raw data of the ELISA test (OD 492 nm values) were converted into a ratio of positive to negative samples (Figure 4/Appendix A). For this purpose, every positive sample was calculated as a ratio to the mean value of a group of uninfected individuals (negative), which was determined with a specific antigen in the ELISA test. This approach allows a statistical analysis of ratios between the individual antigens, with higher ratio values indicating a greater discrepancy between positive and negative results in the raw test value and consequently a separation of seropositive and seronegative results. Higher values can suggest a lower probability of false-negative test results.

A one-way ANOVA with a Bonferroni post hoc test was performed to compare the ratio values for TLA as a reference for the tested antigens. The analysis also compared pairs of chimeric antigens containing the same third antigen in the form of a longer and a shorter amino acid sequence, such as SS-AMA1 and SS-AMA1S (Figure 4). The analyses were performed for each of the positive sample groups, for the chronic phase samples only, or for all positive samples. The detailed results of the analyses are shown in Appendix A. In the acute phase of infection, proteins such as the SS-GRA1, -GRA7, -MIC1S, -MIC3, -P35S, or SAG1-SAG2 showed significantly higher ratio values than TLA. In contrast, the ratio values of the other antigens were significantly lower, with the exception of SS-GRA5, -GRA6, -GRA9, -MAG1S, -MAG1, -MIC1, and -P35 whose ratio values did not differ significantly from TLA.

The results of the analysis show slight deviations when differentiated according to the various phases of a chronic infection. No significant differences were found for SS-AMA1, -GRA5, -MAG1S, and -P35S compared to TLA when the samples were categorized into IgG titer groups >200 IU/mL, 101–200 IU/mL, and <100 IU/mL. However, certain antigens, including SS-GRA1, -GRA9, -MIC3, and SAG1-SAG2, showed significantly elevated ratios in each of these phases compared to TLA. The summary of all chronic samples shows that antigens such as SS-GRA1, -GRA7, -GRA9, -MAG1, -MIC1S, -MIC3, -P35S, or SAG1-SAG2 continue to have higher ratio values compared to TLA. The sum of all positive groups showed minimal changes compared to the chronic group summary, with the SS-P35 antigen now having a higher average ratio value than TLA.

Pairwise analysis in the acute phase showed significantly lower ratios for the shorter GRA5S and MAG1S antigens than for their longer counterparts, and conversely higher ratios for the shorter MIC1S and P35S antigens. The ratios for AMA1 and AMA1S do not differ from each other. This is consistent with the fact that AMA1 and AMA1S do not show significant differences in the chronic phase of infection, with the exception of the chronic group with IgG < 100 IU/mL where SS-AMA1S has a lower ratio compared to SS-AMA1. This discrepancy is statistically significant for all chronic and positive cases. The SS-GRA5S antigen has a significantly lower ratio compared to SS-GRA5 in all groups, with the exception of the group with the high IgG titre > 200 IU/mL. A similar relationship is observed between SS-MAG1 and SS-MAG1S, with a significant difference between the ratios in each group. In contrast, the results for SS-MIC1 and SS-MIC1S show an opposite trend, with SS-MIC1S having a significantly higher ratio in the acute and chronic phase groups, again with the exception of the IgG group with a titre of >200 IU/mL. A similar effect is observed between the average ratio values of SS-P35 and SS-P35S, which are not consistent. In the case of the acute phase, chronic IgG 101–200 IU/mL, and IgG < 100 IU/mL a higher ratio is observed for SS-P35S, while in the IgG > 200 IU/mL group SS-P35S has a significantly lower ratio compared to SS-P35. Consequently, when analyzing the overall data, which includes both chronic and acute individuals, no statistically significant differences were found between these antigens.

The data converted to the positive/negative ratio was also used to compare all trivalent antigens with the SAG1-SAG2 antigen using the same method (Appendix A). It is clear that there are particularly significant differences in the ratios in the acute toxoplasmosis phase group, with the SAG1-SAG2 ratio significantly higher than for all other antigens. This observation is repeated in the chronic phase groups, with the exception of the SS-GRA1 antigen which has a higher ratio value in the IgG > 200 IU/mL, IgG 101–200 IU/mL, and IgG < 100 IU/mL groups. This shows statistical significance in the summary of all samples of the chronic phase. It should be noted that the antigens SS-GRA7, -GRA9, -MIC1S, and SS-MIC3 have a lower but comparable ratio to SAG1-SAG2.

## 3. Discussion

The complex epidemiology of toxoplasmosis, with its multiple routes of transmission and wide range of potential hosts, poses a challenge to the development of effective diagnostic algorithms for both humans and animals. The lack of comprehensive surveillance of parasitic diseases, including toxoplasmosis, in livestock in many countries is largely due to the absence of legislation and the lack of inexpensive and reliable diagnostic tests for routine use on a large scale [37,38]. To address this gap, ongoing research is focusing on the development of new diagnostic tools that could facilitate the control of *T. gondii* infection in livestock. This includes efforts to improve the sensitivity, specificity, and cost-effectiveness of diagnostic tests and to expand the range of diagnostic methods available. In contrast, the epidemiological situation of toxoplasmosis in humans is being monitored more closely, with organizations such as the World Health Organization (WHO) playing a key role in coordinating surveillance efforts. *T. gondii* is globally recognized as a major foodborne pathogen, as reported by authoritative bodies such as the WHO, the Food and Agriculture Organization (FAO), and the European Food Safety Authority (EFSA) [39]. In an FAO/WHO report *T. gondii* was recognized as the fourth-most-important parasite worldwide due to its significant impact on public health. The EFSA categorized *Toxoplasma* as a category III zoonotic agent, highlighting that it should be monitored and placing it alongside other important pathogens such as *Campylobacter* and *Yersinia* [40]. This categorization underlines the importance of monitoring and controlling the spread of *Toxoplasma* to ensure food safety and public health. In terms of disease burden, expressed in disability-adjusted life years (DALYs), *T. gondii* ranks prominently among foodborne pathogens. In the United States, it is the second most common of fourteen foodborne pathogens, and in the Netherlands, it ranks first in terms of DALYs [41]. These statistics highlight the significant impact of *Toxoplasma* infection on human health and emphasize the importance of effective surveillance, prevention, and control measures to contain the spread and associated health risks.

Over the past decades numerous studies have contributed to a better understanding of the antigenic structure and function of *T. gondii* antigens, providing the basis for the development of new diagnostic tools. Ongoing research to develop novel diagnostic tools based on recombinant *T. gondii* proteins is promising to improve the detection and control of toxoplasmosis in both humans [19] and animals [42]. To facilitate standardization and minimize the costs associated with the need to produce and purify each protein individually, research began on recombinant chimeric proteins, which represent a new group of protein preparations. To obtain recombinant chimeric proteins, fragments of two or more genes encoding individual single antigens are combined to form a so-called fusion gene. By expressing the fusion gene, a recombinant chimeric protein is produced, which should be able to interact with antibodies directed against individual antigens of the parasite.

The purpose of this study was to determine the diagnostic utility of 18 newly produced recombinant trivalent chimeric proteins for the detection of specific anti-*T. gondii* antibodies in human sera in an indirect IgG ELISA. The recombinant trivalent chimeric proteins obtained consisted of three immunodominant fragments of well-known and characterized parasite antigens. Each recombinant chimeric protein contained an immunodominant core fragment of the parasite surface antigens SAG1 and SAG2 and an additional immunodominant fragment of one of the parasite antigens, such as AMA1, GRA1, GRA2, GRA5, GRA6, GRA7, GRA9, LDH2, MAG1, MIC1, MIC3, P35, and ROP1. In addition, a recombinant divalent chimeric protein SAG1-SAG2, which has already been described in the literature by other research teams, was obtained for comparison purposes [43].

To date, a limited number of studies have demonstrated the usefulness of recombinant chimeric proteins for the detection of specific anti-*T. gondii* antibodies. In one study, two antigens consisting of the most immunoreactive regions of MIC2_157–235_, MIC3_234–307_, SAG1_182–312_ (GST-EC2), and GRA3_36–134_, GRA7_24–102_, M2AP_37–263_ (GST-EC3) were used. The sensitivity of the IgG ELISA for the EC2 and EC3 proteins was estimated to be 100%. These chimeric proteins have also been successfully used to detect IgM antibodies in sera from patients in the early phase of *T. gondii* infection, with EC2 and EC3 antigens showing 98% and 84% reactivity, respectively [44]. Another antigen that was used to detect IgG and IgM immunoglobulins in human sera was the chimeric protein SAG1/2. Western blot assays demonstrated their usefulness in detecting infections in patients with early-acute, acute, and chronic toxoplasmosis [43]. Dai et al. developed a recombinant multiepitope fusion protein (rMEP) consisting of three antigenic determinants, SAG1_309–318_, SAG2_109–118_, and SAG3_347–356_, which are recognized by B lymphocytes. The rMEP protein showed high reactivity in IgG ELISA tests of serum samples from both acute (87.5%) and chronic (97.4%) *T. gondii* infections [45]. The capacity to distinguish the acute from the chronic phase of *T. gondii* infection was confirmed in another study, which showed that this ability depends on the choice of the optimal concentration of the rMEP antigen for IgG ELISA testing. The sensitivity of the IgG ELISA was estimated to be 25.9% for sera from the acute phase of *T. gondii* infection and 97.1% for sera from the chronic phase of the disease. However, the authors did not report the number of sera used to determine the cut-off value, and a relatively small pool of serum samples was used to determine the specificity of the test [46]. Other studies in which three divalent recombinant chimeric proteins, such as P35-MAG1, MIC1-ROP1, and MAG1-ROP1, were obtained showed that these protein preparations allowed the development of IgG ELISA tests with 100% specificity and a sensitivity of 45.8%, 47%, and 47%, respectively. However, they may allow differentiation between the acute and chronic phases based on the testing of IgG antibodies only. The reactivity of P35-MAG1, MIC1-ROP1, and MAG1-ROP1 protein preparations with anti-*T. gondii* IgG antibodies in serum samples from patients with the acute phase of infection was 100%, 77.3%, and 86.4%, respectively. In comparison, the reactivity with IgG antibodies in sera from the chronic phase of infection was 26.2%, 36.1%, and 32.8%, respectively. The IgM ELISA test showed that the reactivity of the protein preparations P35-MAG1, MIC1-ROP1, and MAG1-ROP1 was 81.8%, 72.7%, and 59.1%, respectively. Unfortunately, false-positive results were also obtained in sera from patients in the chronic phase of infection (IgM negative, IgG positive, high IgG avidity index), as well as in uninfected individuals (a serum sample of 20 sera served as a control) [47]. In our previous work we also demonstrated the diagnostic utility of recombinant chimeric proteins, such as MIC1-MAG1 [48], MIC1-MAG1-SAG1 [49], SAG2-GRA1-ROP1S, and SAG2-GRA1-ROP1L [50]. In these studies, it was demonstrated that individual recombinant proteins or mixtures of recombinant proteins have a lower reactivity than the corresponding recombinant chimeric proteins and the polyvalent native antigen TLA. IgG ELISA tests based on the recombinant chimeric proteins MIC1-MAG1, MIC1-MAG1-SAG1, SAG2-GRA1-ROP1_S_, and SAG2-GRA1-ROP1_L_ were characterized by a 100% specificity and a sensitivity of 90.9%, 98.1%, 88.4%, and 100%, respectively. In addition, the results for protein preparations of SAG2-GRA1-ROP1_S_ and SAG2-GRA1-ROP1_L_ showed that the size of the immunodominant fragment in the recombinant chimeric protein can determine its reactivity with specific anti-*T. gondii* antibodies. In our recent work, we have performed comprehensive studies on the diagnostic potential of recombinant tetravalent chimeric proteins consisting of immunodominant fragments of the SAG2, GRA1, and ROP1 antigens and fragments of the AMA1 antigen in different sizes [51]. Four recombinant tetravalent chimeric proteins, such as SAG2-GRA1-ROP1-AMA1N, AMA1N-SAG2-GRA1-ROP1, AMA1C-SAG2-GRA1-ROP1, and AMA1-SAG2-GRA1-ROP1, were obtained and tested by IgG ELISA, IgM ELISA, and IgG avidity ELISA. All protein preparations were characterized by a high reactivity to specific anti-*T. gondii* IgM and IgG antibodies. Among the protein preparations obtained, AMA1-SAG2-GRA1-ROP1 gave similar results to the TLA-based tests. This protein preparation enabled the development of IgG ELISA and IgG avidity ELISA tests with a specificity and sensitivity of 100%, and the development of an IgM ELISA test with a sensitivity of 95.5% and a specificity of 99%. These results show that the recombinant tetravalent chimeric protein AMA1-SAG2-GRA1-ROP1 has the potential to distinguish specific antibodies from serum samples from acute and chronic phases of *T. gondii* infection and can be an alternative to TLA.

The results of this study confirm that recombinant chimeric proteins composed of different immunodominant fragments of parasite antigens can be an excellent alternative to TLA, which is commonly used in commercial assays. All tested antigen compositions allowed the detection of IgG antibodies in human sera with 100% specificity and a sensitivity of 98.5–100%. In fact, of the 18 trivalent proteins used in the study, only three had a sensitivity of less than 100%. It has been shown again that the size of the immunodominant fragment of a given antigen can determine its ability to interact with specific antibodies, which is related to the number of epitopes, as we mentioned in our previous work on recombinant chimeric proteins with different sized immunodominant fragments of the AMA1 antigen [51]. The results of our studies of the same protein preparations in IgG ELISA assays which used sera from small ruminants showed that recombinant chimeric proteins, such as SAG1-SAG2-GRA5, SAG1-SAG2-GRA9, SAG1-SAG2-MIC1, SAG1-SAG2-MIC3, SAG1-SAG2-P35, and SAG1-SAG2-ROP1, allowed the development of assays with 100% sensitivity and specificity [36]. In addition, five of these preparations, apart from SAG1-SAG2-ROP1, also showed 100% specificity for human IgG. This observation is particularly important as it demonstrates the potential diagnostic utility of these proteins for various *T. gondii* intermediate host species, allowing us to be optimistic about the possibility of replacing the currently used mixture of native antigens with recombinant proteins in the serodiagnosis of *T. gondii* infections in humans and other animals.

Taking into account the results of previous work and the research presented in this work, recombinant chimeric proteins have proven to be promising tools for the serodiagnosis of toxoplasmosis, offering several advantages over conventional TLA-based diagnostic methods. In addition, recombinant chimeric proteins possess significantly greater diagnostic potential than single recombinant proteins or their mixtures. Recombinant chimeric proteins combine multiple antigenic regions in a single construct, which increases their ability to capture a broad spectrum of specific antibodies in patient serum samples. This comprehensive approach can improve the accuracy of serological tests by minimizing the risk of false-positive and false-negative results. In addition, the specific arrangement of antigenic regions within the recombinant chimeric protein construct can improve specificity by reducing cross-reactivity with antibodies against other pathogens. Due to the precise control of protein composition and quality, recombinant chimeric proteins offer greater standardization and reproducibility compared to mixtures of recombinant proteins or native antigen preparations, such as TLA. Some recombinant chimeric proteins have shown promise in differentiating between acute and chronic phases of toxoplasmosis as they are able to detect IgG antibodies in serum samples at a specific phase of *T. gondii* infection [46,47]. Recombinant chimeric proteins have the potential to be developed into rapid diagnostic tests suitable for point-of-care situations due to their stability, ease of production, and compatibility with various test formats. Considering that the tests currently used for the diagnosis of *T. gondii* infections must be performed in a specialized diagnostic laboratory that has the appropriate equipment to perform the test, new rapid tests would speed up the diagnosis of parasite infection and the initiation of appropriate drug therapy. This is particularly important in the case of risk groups such as pregnant women or immunocompromised patients. Research must continue to optimize the design and performance of recombinant chimeric proteins for the diagnosis of toxoplasmosis. This includes further validation studies with different patient groups, research into additional antigen combinations, and the development of novel assay platforms for broad clinical use.

## 4. Materials and Methods

### 4.1. Parasite Culture and Preparation of the Toxoplasma Lysate Antigen (TLA)

The *T. gondii* RH strain (PRA-310, ATCC^®^, Manassas, VA, USA) was maintained *in vitro* on human Hs27 fibroblasts (Hs27, CRL-1634, ATCC^®^, Manassas, VA, USA) and used to produce TLA according to the previously described protocol [36].

### 4.2. T. gondii RNA Isolation and cDNA Synthesis

Total RNA was isolated from freshly harvested tachyzoites using a Total RNA Mini Plus D kit (A&A Biotechnology, Gdynia, Poland) according to the manufacturer’s procedure. Using a TranScriba Kit (A&A Biotechnology, Gdynia, Poland) the cDNA single strand was synthesized according to the manufacturer’s instructions and stored at −20 °C.

### 4.3. Construction of the Recombinant Plasmids

For all 19 recombinant chimeric proteins used in this study, those previously described [36], and the newly constructed SS-GRA5S, the DNA sequences of the genes encoding the corresponding antigens of *T. gondii* were retrieved from GenBank. The nucleotide sequences (GenBank Accession No.) and amino acid sequences (GenPept Accession No.) of all *T. gondii* antigens used for the construction of recombinant chimeric proteins are summarized in Appendix A. The cDNA obtained from the *T. gondii* RH strain served as a template for the amplification of genes encoding immunodominant fragments of antigens using a standard PCR amplification protocol with Phusion High-Fidelity DNA Polymerase (Thermo Fisher Scientific, Inc., Waltham, MA, USA). The primers used to amplify the DNA of the individual gene fragments were ordered from Genomed S.A. (Warsaw, Poland) and are listed in Appendix A. In the case of the recombinant plasmid encoding the recombinant chimeric protein SAG1-SAG2-GRA5S, the PCR products were inserted into the BglII site of the expression vector pET30 Ek/LIC (Merck, KGaA, Darmstadt, Germany). The primers for the cloning reactions have designed to allow the use of a In-Fusion HD Cloning Kit (Takara Bio Inc., Kasatsu, Shiga, Japan), which is based on DNA recombination.

### 4.4. Production and Purification of Recombinant Proteins

Again, for all proteins tested in the study, including the newly developed SS-GRA5S antigen, the *E. coli* strain Rosetta(DE3)pLysS transformed with recombinant plasmids was cultivated overnight at 30 °C in Terrific Broth (TB) medium supplemented with 20 µg/mL kanamycin and 34 µg/mL chloramphenicol. Protein expression was performed as previously described [36]. The protein extracts, obtained by dissolving the inclusion bodies with 5 M urea, were then purified using a Ni Sepharose™ 6 Fast Flow column (Cytiva, Little Chalfont, England, UK) according to the manufacturer’s protocol. The starting buffer for suspension and subsequent sonication of the bacterial cells was a buffer with the following composition: 5 M urea, 0.5 M NaCl, 20 mM Tris, 5 mM imidazole, and 0.1% Triton X-100. The recombinant chimeric proteins were purified in buffers with a pH of 7.9, with the exception of the recombinant chimeric protein SAG1-SAG2-P35 whose isoelectric point was much higher and so buffers with a pH of 9.5 had to be used. After sonication and centrifugation, the clear lysates were loaded onto purification columns and the columns were washed with buffers containing increasing concentrations of imidazole to elute other proteins. Finally, the recombinant chimeric proteins were eluted with buffer without denaturant at an imidazole concentration of 0.5 M. With the developed method for purifying recombinant chimeric proteins, antigen preparations with an electrophoretic purity of over 90% were obtained.

### 4.5. Human Serum Samples

Human serum samples from the collection of the Department of Tropical Medicine and Parasitology of the Medical University of Gdańsk, Poland, were used for the study, as were fully anonymized samples of biological material collected from patients of the University Center for Maritime and Tropical Medicine as part of routine laboratory diagnostics, with patients providing consent for use for scientific purposes. All procedures performed in this study were in accordance with the ethical standards of the institutional research committee and the 1964 Declaration of Helsinki and its subsequent amendments or comparable ethical standards.

Serum samples were categorized as positive or negative based on specific IgM and IgG antibody levels according to the manual of the commercial diagnostic test used (VIDAS TOXO IgM and VIDAS TOXO IgG II; bioMérieux, Marcy l’Etoile, France). In addition, all seropositive serum samples were tested for their IgG avidity index using a VIDAS TOXO IgG AVIDITY (bioMérieux, Marcy l’Etoile, France). The pool of 338 serum samples were categorized into appropriate groups and subgroups according to the commercial test results, as shown in Table 3.

### 4.6. IgG ELISA—Human Serum Samples

The usefulness of the tested recombinant chimeric proteins for the detection of specific anti-*T. gondii* antibodies in human serum samples were evaluated using an indirect ELISA test, as previously described [51]. MaxiSorp multiwell plates (Thermo Fisher Scientific, Inc., Waltham, MA, USA) were coated overnight with recombinant chimeric protein at a final concentration of 2.5 µg/mL and 1 µg/mL for TLA in a coating carbonate buffer, which had a pH 9.6 (Merck, KGaA, Darmstadt, Germany). A solution of 1% bovine serum albumin (Merck, KGaA, Darmstadt, Germany) and 0.05% Tween 20 in PBS was used to block and dilute the primary and secondary antibodies. All tested human serum samples were diluted 1:100. The secondary goat anti-human IgG antibodies conjugated with horseradish peroxidase (HRP) (Jackson ImmunoResearch Europe Ltd., Cambridgeshire, UK) were used at a dilution of 1:32,000. The colour reaction was developed using a chromogenic o-phenylenediamine dihydrochloride substrate (Merck, KGaA, Darmstadt, Germany). After 45 min of incubation in the dark, the reaction was stopped by adding 2 M sulphuric acid, and the OD was measured at 492 nm (Multiskan FC; Thermo Fisher Scientific, Inc., Waltham, MA, USA). The reactivity of the serum samples with the individual protein preparation was considered positive if the OD value was above the cut-off value (OD + 2 standard deviations) which had been determined with serum samples from the control group.

### 4.7. Statistical Analysis

Graphs and statistical analyses were performed using GraphPad Prism 10.3.0 (Dotmatics, GraphPad Software, San Diego, CA, USA). Shapiro–Wilk and D’Agostino–Pearson tests were used to assess the Gaussian distribution of the data and residuals, along with analysis of Q-Q plots. The Brown–Forsythe test was used to test for the equality of group variances. A receiver operating characteristics (ROCs) analysis was performed to obtain the area under the curve (AUC), sensitivity, and specificity of the test. Pearson correlation coefficient analysis was used to assess the correlation between the recombinant chimeric proteins vs. TLA results for each serum sample. Repeated measures (RMs) one-way ANOVA was used to compare the ratios of positive/negative human serum samples for each antigen used, followed by a Bonferroni multiple comparison test to compare each antigen with TLA and to compare the antigens with the short and long versions of the end antigens of the trivalent chimeric antigen (e.g., SS-AMA1 vs. SSAMA1S). The ratio is calculated by dividing the value of a given sample by the average test value of samples from uninfected individuals for each antigen used in the test. The alpha was set to 0.05 for all statistical analyses.

## 5. Conclusions

In summary, recombinant chimeric proteins represent a promising avenue for improving toxoplasmosis serodiagnosis, offering enhanced sensitivity, specificity, and standardization compared to conventional methods. With ongoing advances in antigen design, assay technology, and clinical validation, these innovative reagents have the potential to significantly impact the diagnosis and treatment of *T. gondii* infections.

## Figures and Tables

**Figure 1 ijms-26-05621-f001:**
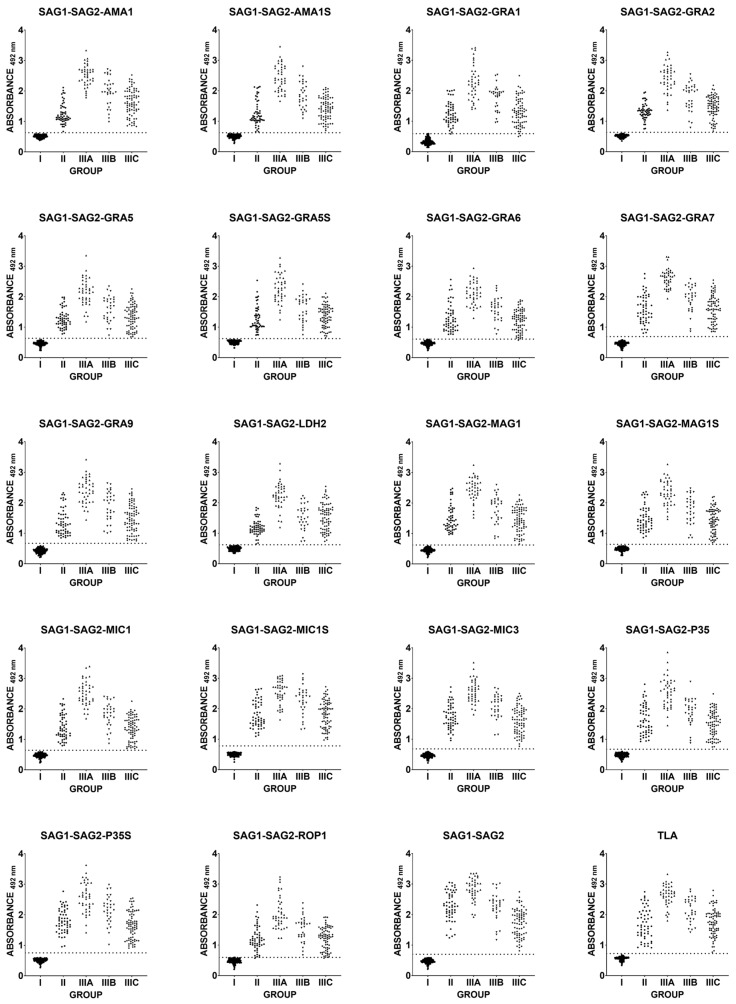
Comparison of immunoreactivity in IgG ELISA using recombinant chimeric proteins and TLA, with different groups of infected individuals. Group: I—uninfected, II—suspected acute, IIIA—chronic, IgG > 200 IU/mL, IIIB—chronic, IgG 101–200 IU/mL, and IIIC—chronic, IgG < 100 IU/mL. The dotted line corresponds to the ROC cut-off.

**Figure 2 ijms-26-05621-f002:**
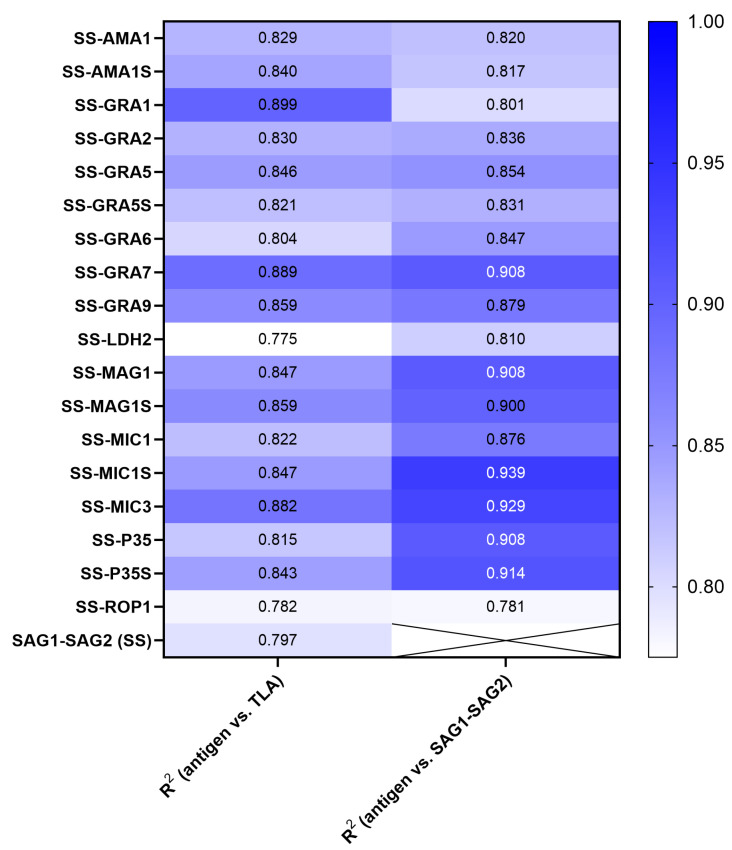
Heatmap of R^2^ values from the Pearson correlation analysis between TLA and recombinant antigens values for each individual, or SAG1-SAG2 antigen values vs. trivalent recombinant antigens values.

**Figure 3 ijms-26-05621-f003:**
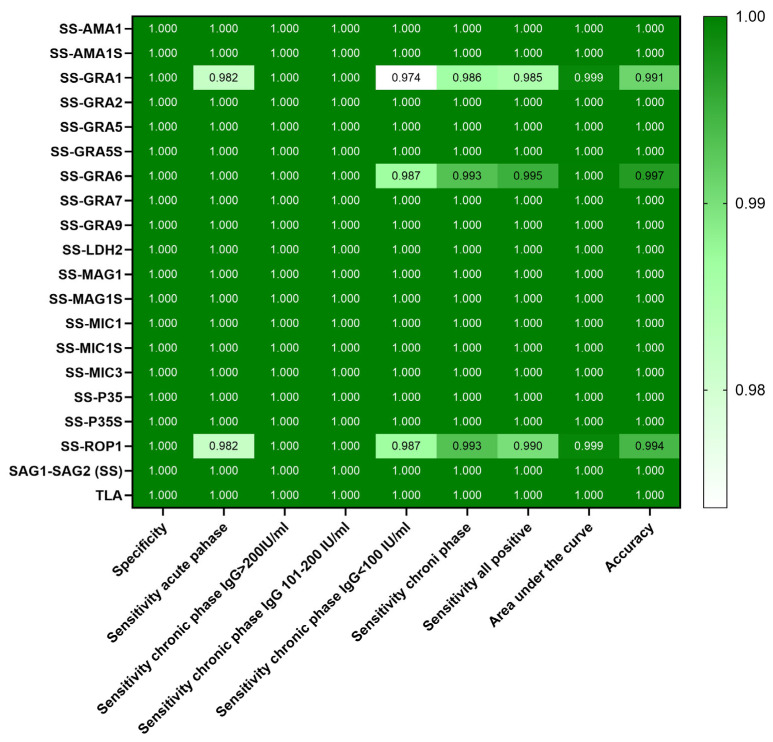
Heatmap of specificity, sensitivity, AUC, and accuracy values from ROC analysis for each antigen.

**Figure 4 ijms-26-05621-f004:**
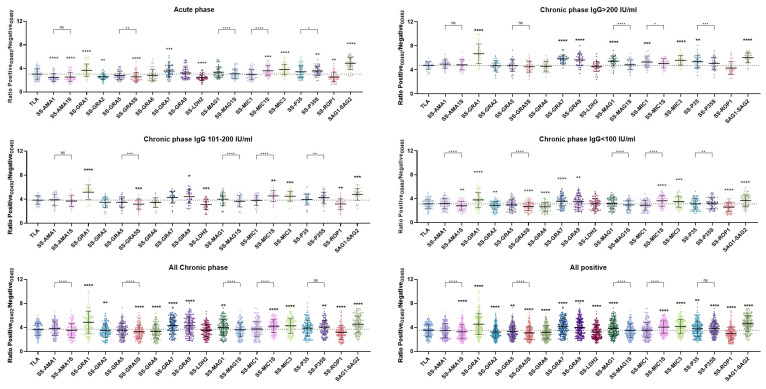
Comparison of immunoreactivity in IgG ELISA using a positive/negative ratio. The ratio is calculated by dividing the value of a given sample by the average value of samples from noninfected people for the antigen used in the test. Analysis was performed using repeated measures one-way ANOVA followed by Bonferroni’s multiple comparison test to compare each antigen with TLA and to compare antigens with the short and long version of end antigens (brackets). ns—*p*_adj_ ≥ 0.05, (*)—*p*_adj_ < 0.05, (**)—*p*_adj_ ≤ 0.01, (***)—*p*_adj_ ≤ 0.001, and (****)—*p*_adj_ ≤ 0.0001. Data presented as mean and standard deviation. The dotted line indicates the mean of TLA.

**Table 1 ijms-26-05621-t001:** Pearson’s correlation coefficient between the results obtained for the recombinant chimeric proteins vs. the TLA results and the recombinant trivalent chimeric proteins vs. the recombinant divalent chimeric protein SAG1-SAG2.

Recombinant Chimeric Protein	vs.	*r* (95% CI)	*r^2^*	*p* Value
SAG1-SAG2-AMA1	TLA SAG1-SAG2	0.9102 (0.8900–0.9269) 0.9055 (0.8843–0.9230)	0.8285 0.8200	<0.0001 <0.0001
SAG1-SAG2-AMA1S	TLA SAG1-SAG2	0.9163 (0.8973–0.9319) 0.9038 (0.8822–0.9217)	0.8396 0.8169	<0.0001 <0.0001
SAG1-SAG2-GRA1	TLA SAG1-SAG2	0.9481 (0.9361–0.9579) 0.8948 (0.8713–0.9142)	0.8989 0.8007	<0.0001 <0.0001
SAG1-SAG2-GRA2	TLA SAG1-SAG2	0.9112 (0.8911–0.9277) 0.9142 (0.8948–0.9302)	0.8302 0.8358	<0.0001 <0.0001
SAG1-SAG2-GRA5	TLA SAG1-SAG2	0.9198 (0.9016–0.9348) 0.9241 (0.9068–0.9383)	0.8460 0.8540	<0.0001 <0.0001
SAG1-SAG2-GRA5S	TLA SAG1-SAG2	0.9058 (0.8846–0.9233) 0.9117 (0.8918–0.9281)	0.8205 0.8312	<0.0001 <0.0001
SAG1-SAG2-GRA6	TLA SAG1-SAG2	0.8967 (0.8735–0.9157) 0.9202 (0.9021–0.9351)	0.8040 0.8467	<0.0001 <0.0001
SAG1-SAG2-GRA7	TLA SAG1-SAG2	0.9428 (0.9296–0.9536) 0.9529 (0.9419–0.9618)	0.8889 0.9079	<0.0001 <0.0001
SAG1-SAG2-GRA9	TLA SAG1-SAG2	0.9267 (0.9100–0.9404) 0.9373 (0.9229–0.9490)	0.8587 0.8785	<0.0001 <0.0001
SAG1-SAG2-LDH2	TLA SAG1-SAG2	0.8804 (0.8539–0.9023) 0.8998 (0.8773–0.9183)	0.7751 0.8096	<0.0001 <0.0001
SAG1-SAG2-MAG1	TLA SAG1-SAG2	0.9202 (0.9021–0.9351) 0.9531 (0.9422–0.9619)	0.8469 0.9083	<0.0001 <0.0001
SAG1-SAG2-MAG1S	TLA SAG1-SAG2	0.9270 (0.9103–0.9407) 0.9485 (0.9366–0.9582)	0.8593 0.8997	<0.0001 <0.0001
SAG1-SAG2-MIC1	TLA SAG1-SAG2	0.9065 (0.8854–0.9238) 0.9358 (0.9211–0.9479)	0.8217 0.8757	<0.0001 <0.0001
SAG1-SAG2-MIC1S	TLA SAG1-SAG2	0.9201 (0.9019–0.9350) 0.9689 (0.9617–0.9748)	0.8466 0.9388	<0.0001 <0.0001
SAG1-SAG2-MIC3	TLA SAG1-SAG2	0.9391 (0.9251–0.9505) 0.9641 (0.9557–0.9709)	0.8819 0.9294	<0.0001 <0.0001
SAG1-SAG2-P35	TLA SAG1-SAG2	0.9026 (0.8807–0.9206) 0.9530 (0.9421–0.9619)	0.8147 0.9083	<0.0001 <0.0001
SAG1-SAG2-P35S	TLA SAG1-SAG2	0.9184 (0.8998–0.9336) 0.9561 (0.9459–0.9644)	0.8434 0.9141	<0.0001 <0.0001
SAG1-SAG2-ROP1	TLA SAG1-SAG2	0.8845 (0.8588–0.9057) 0.8837 (0.8578–0.9050)	0.7823 0.7808	<0.0001 <0.0001
SAG1-SAG2	TLA	0.8928 (0.8688–0.9125)	0.7970	<0.0001

**Table 2 ijms-26-05621-t002:** ROC analysis of the results obtained in the IgG ELISA using human sera.

Recombinant Chimeric Protein	Calculated Cut-Off	ROC Cut-Off	Sensitivity [%] (95% CI)	Specificity [%] (95% CI)	Area Under the Curve (95% CI)	*p* Value
SAG1-SAG2-AMA1	0.6083	0.6320	100 (98.12–100)	100 (97.29–100)	1.000 (1.000–1.000)	<0.0001
SAG1-SAG2-AMA1S	0.6169	0.6233	100 (98.12–100)	100 (97.29–100)	1.000 (1.000–1.000)	<0.0001
SAG1-SAG2-GRA1	0.5445	0.5918	98.5 (95.68–99.59)	100 (97.29–100)	0.9992 (0.9981–1.000)	<0.0001
SAG1-SAG2-GRA2	0.6113	0.6363	100 (98.12–100)	100 (97.29–100)	1.000 (1.000–1.000)	<0.0001
SAG1-SAG2-GRA5	0.5958	0.6350	100 (98.12–100)	100 (97.29–100)	1.000 (1.000–1.000)	<0.0001
SAG1-SAG2-GRA5S	0.6149	0.6240	100 (98.12–100)	100 (97.29–100)	1.000 (1.000–1.000)	<0.0001
SAG1-SAG2-GRA6	0.6015	0.6075	99.5 (97.22–99.97)	100 (97.29–100)	0.9999 (0.9995–1.000)	<0.0001
SAG1-SAG2-GRA7	0.5948	0.6908	100 (98.12–100)	100 (97.29–100)	1.000 (1.000–1.000)	<0.0001
SAG1-SAG2-GRA9	0.5898	0.6695	100 (98.12–100)	100 (97.29–100)	1.000 (1.000–1.000)	<0.0001
SAG1-SAG2-LDH2	0.6134	0.6233	100 (98.12–100)	100 (97.29–100)	1.000 (1.000–1.000)	<0.0001
SAG1-SAG2-MAG1	0.5911	0.6233	100 (98.12–100)	100 (97.29–100)	1.000 (1.000–1.000)	<0.0001
SAG1-SAG2-MAG1S	0.6112	0.6380	100 (98.12–100)	100 (97.29–100)	1.000 (1.000–1.000)	<0.0001
SAG1-SAG2-MIC1	0.6080	0.6433	100 (98.12–100)	100 (97.29–100)	1.000 (1.000–1.000)	<0.0001
SAG1-SAG2-MIC1S	0.6089	0.7798	100 (98.12–100)	100 (97.29–100)	1.000 (1.000–1.000)	<0.0001
SAG1-SAG2-MIC3	0.5967	0.6813	100 (98.12–100)	100 (97.29–100)	1.000 (1.000–1.000)	<0.0001
SAG1-SAG2-P35	0.6144	0.6738	100 (98.12–100)	100 (97.29–100)	1.000 (1.000–1.000)	<0.0001
SAG1-SAG2-P35S	0.6246	0.7523	100 (98.12–100)	100 (97.29–100)	1.000 (1.000–1.000)	<0.0001
SAG1-SAG2-ROP1	0.6184	0.5985	99 (96.43–99.82)	100 (97.29–100)	0.9993 (0.9982–1.000)	<0.0001
SAG1-SAG2	0.6064	0.6995	100 (98.12–100)	100 (97.29–100)	1.000 (1.000–1.000)	<0.0001
TLA	0.6804	0.7245	100 (98.12–100)	100 (97.29–100)	1.000 (1.000–1.000)	<0.0001

**Table 3 ijms-26-05621-t003:** Groups of human serum samples.

Group of Serum Samples		Phase of *T. gondii* Infection	Number of Serum Samples	IgM	IgG	IgG Avidity
I		uninfected	138	−	−	not tested
II		suspected acute	54	+	+	low
III	IIIA	chronic	40	+/−	>200 IU/mL	high
	IIIB		30	−	101–200 IU/mL	
	IIIC		76	−	≤100 IU/mL	
	IIIA-C		146	+/−	+	

## Data Availability

All data generated and analyzed during this study that support the findings are included in this published article/Appendix A. Further inquiries can be directed to the corresponding author.

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
