# Peer review of "The Trivalent Recombinant Chimeric Proteins Containing Immunodominant Fragments of Toxoplasma gondii SAG1 and SAG2 Antigens in Their Core—A Good Diagnostic Tool for Detecting IgG Antibodies in Human Serum Samples"

_ijms, 2025, doi:10.3390/ijms26125621_

Round 1
Reviewer 1 Report
Comments and Suggestions for Authors
The authors evaluated toxoplasma recombinant chimeric proteins. A chimeric antigen consisting of surface antigens SAG1-SAG2 was selected. Further research will be conducted based on this, 18 trivalent recombinant chimeric proteins containing immunodominant fragments of these two antigens. The indirect ELISA test show they are extremely sensitive to IgG in human serum samples.
All the data of this paper is clear and easy to understand. The background information provided is detailed, which is the strength of this article. I only have one suggestion:The authors should provide gene IDs for all proteins (such as NCBI's Gene ID, ToxoDB ID, etc.). It is convenient for readers to find detailed information and the reproducibility and verifiability of supporting data
Author Response
Thank you very much for taking the time to review this manuscript. Please find the detailed response below and the corrections highlighted in the re-submitted files.
Comments 1: All the data of this paper is clear and easy to understand. The background information provided is detailed, which is the strength of this article.
Response 1: Thank you very much for appreciating our work.
Comments 2: I only have one suggestion:The authors should provide gene IDs for all proteins (such as NCBI's Gene ID, ToxoDB ID, etc.). It is convenient for readers to find detailed information and the reproducibility and verifiability of supporting data.
Response 2: Thank you for pointing this out. We agree with this comment. We didn't want to repeat the information that was already included in the work in which we described in detail the construction of recombinant chimeric proteins Ferra, B.T. et. al. The Development of Toxoplasma gondii Recombinant Trivalent Chimeric Proteins as an Alternative to Toxoplasma Lysate Antigen (TLA) in Enzyme-Linked Immunosorbent Assay (ELISA) for the Detection of Immunoglobulin G (IgG) in Small Ruminants. Int. J. Mol. Sci. 2024, 25, 4384. https://doi.org/10.3390/ijms25084384. However, as suggested, we added a Table S7 to the supplementary materials in which we placed the nucleotide sequences, GenBank Accession No. and amino acid sequences, GenPept Accession No. of the antigens. We have included this information in the Materials and Methods section. We have marked it in yellow in the revised version of the manuscript that was submitted.
Reviewer 2 Report
Comments and Suggestions for Authors
Reviewing the paper with the title “The trivalent recombinant chimeric proteins containing in their 2 core immunodominant fragments of Toxoplasma gondii SAG1 3 and SAG2 antigens – a good diagnostic tool for detecting IgG 4 antibodies in human serum samples”
Strength
Comprehensive testing: 19 different recombinant antigens and test across 338 human samples
Point to improve: Data Presentation and Figures
While I appreciate the thoroughness of the experimental work, especially the testing of 19 chimeric proteins across a large serum cohort. I found the manuscript's data presentation a bit too compressed. The study includes only two main figures, which makes it challenging to appreciate the depth and nuance of the dataset. Given the number of antigens tested and the breakdown by infection phase (acute vs. various chronic states), I believe the paper would benefit from additional figures to more clearly illustrate key findings.
For example, including individual ELISA heatmaps comparing sensitivity/specificity across all constructs could help the reader quickly assess which antigens perform best under which conditions.
In short, the intentions of the study are strong and will benefit society, but expanding the visual presentation would make the story more accessible and impactful.
Author Response
Thank you very much for taking the time to review this manuscript. Please find the detailed response below and the corrections highlighted in the re-submitted files.
Comment 1: While I appreciate the thoroughness of the experimental work, especially the testing of 19 chimeric proteins across a large serum cohort. I found the manuscript's data presentation a bit too compressed. The study includes only two main figures, which makes it challenging to appreciate the depth and nuance of the dataset. Given the number of antigens tested and the breakdown by infection phase (acute vs. various chronic states), I believe the paper would benefit from additional figures to more clearly illustrate key findings. For example, including individual ELISA heatmaps comparing sensitivity/specificity across all constructs could help the reader quickly assess which antigens perform best under which conditions. In short, the intentions of the study are strong and will benefit society, but expanding the visual presentation would make the story more accessible and impactful.
Response 1: First of all, thank you very much for appreciating our work. We agree with this comment. We didn't want to multiply graphs showing the same results, and we didn't think that heatmaps could be used to summarize our results. Thank you very much for this valuable suggestion. We prepared two heatmaps (Figure 2 and Figure 3) that we included in the main manuscript and two additional figures for the supplementary materials (Figures S2 and Figure S3). It was necessary to make changes to the figure numbers. All changes introduced in the manuscript are marked in green.
Round 2
Reviewer 2 Report
Comments and Suggestions for Authors
-
Author Response
no more comments received